# The S100B Protein: A Multifaceted Pathogenic Factor More Than a Biomarker

**DOI:** 10.3390/ijms24119605

**Published:** 2023-05-31

**Authors:** Fabrizio Michetti, Maria Elisabetta Clementi, Rosa Di Liddo, Federica Valeriani, Francesco Ria, Mario Rende, Gabriele Di Sante, Vincenzo Romano Spica

**Affiliations:** 1Department of Neuroscience, Catholic University of the Sacred Heart, 00168 Rome, Italy; 2IRCCS San Raffaele Scientific Institute, Università Vita-Salute San Raffaele, 20132 Milan, Italy; 3Department of Medicine, LUM University, 70010 Casamassima, Italy; 4Genes, Via Venti Settembre 118, 00187 Roma, Italy; 5Istituto di Scienze e Tecnologie Chimiche “Giulio Natta” SCITEC-CNR, 00168 Rome, Italy; elisabetta.clementi@scitec.cnr.it; 6Department of Pharmaceutical and Pharmacological Sciences, University of Padova, 35131 Padova, Italy; rosa.diliddo@unipd.it; 7Laboratory of Epidemiology and Biotechnologies, Department of Movement, Human and Health Sciences, University of Rome “Foro Italico”, 00135 Rome, Italy; federica.valeriani@uniroma4.it (F.V.); vincenzo.romanospica@uniroma4.it (V.R.S.); 8Department of Translational Medicine and Surgery, Section of General Pathology, Catholic University of the Sacred Heart, 00168 Rome, Italy; francesco.ria@unicatt.i; 9Department of Medicine and Surgery, Section of Human, Clinical and Forensic Anatomy, University of Perugia, 06132 Perugia, Italy; mario.rende@unipg.it (M.R.); gabriele.disante@unipg.it (G.D.S.)

**Keywords:** S100B protein, pathogenic factor

## Abstract

S100B is a calcium-binding protein mainly concentrated in astrocytes in the nervous system. Its levels in biological fluids are recognized as a reliable biomarker of active neural distress, and more recently, mounting evidence points to S100B as a Damage-Associated Molecular Pattern molecule, which, at high concentration, triggers tissue reactions to damage. S100B levels and/or distribution in the nervous tissue of patients and/or experimental models of different neural disorders, for which the protein is used as a biomarker, are directly related to the progress of the disease. In addition, in experimental models of diseases such as Alzheimer’s and Parkinson’s diseases, amyotrophic lateral sclerosis, multiple sclerosis, traumatic and vascular acute neural injury, epilepsy, and inflammatory bowel disease, alteration of S100B levels correlates with the occurrence of clinical and/or toxic parameters. In general, overexpression/administration of S100B worsens the clinical presentation, whereas deletion/inactivation of the protein contributes to the amelioration of the symptoms. Thus, the S100B protein may be proposed as a common pathogenic factor in different disorders, sharing different symptoms and etiologies but appearing to share some common pathogenic processes reasonably attributable to neuroinflammation.

## 1. Introduction

The term S100B refers to a protein identified in the mid-1960s from brain extracts to compare proteins identified in the nervous tissue with those localized in other tissues. It is characterized by an intriguing, unusual solubility in a 100% saturated solution with ammonium sulfate [1], and this characteristic was the basis of its denomination, which was originally S100 protein. At present, the S100 protein family comprises more than 20 calcium-binding proteins, mostly formed by two identical peptides (homodimers), exhibiting structural similarities with different degrees of amino acid homology, located in different tissues where they modulate the activity of many targets [2]. S100B is an acidic homodimer of 9–14 kDa per peptide (monomer) and constitutes the bulk of the protein fraction that was originally isolated from brain extracts; thus, it has been regarded as being specific for this tissue for approximately two decades, but later was shown also to be present in definite extra-neural cell types, as indicated below. Interestingly, the amino acid composition and conformation of S100B, as for other proteins of the S100 family, are highly conserved in vertebrate species, suggesting that it may have a crucially conserved biological role(s), although they have not been identified. In the nervous system, although S100B is mainly concentrated in astrocytes, it is also expressed in other glial cell types, such as oligodendrocytes, Schwann cells, ependymal cells, retinal Müller cells, and enteric glial cells. The protein has even been reported to be contained in specific neuronal subpopulations in the brainstem and some ganglionic peripheral cells. In addition, as above indicated, S100B has also been detected in definite non-neural cell types, such as chondrocytes, melanocytes, Langerhans cells, dendritic cells of lymphoid organs, some lymphocyte cell types, adrenal medulla satellite cells, skeletal muscle satellite cells, tubular kidney cells, non-nervous structures of the eye, such as corneal endothelial cells and lens, iris, ciliary body epithelial cells, Leydig cells, and adipocytes, which constitute a site of concentration for the protein comparable to astrocytes [3]. While the functional characteristics and cell localization of neural S100B have been extensively studied, its properties in non-neural locations have received poor attention, although they would reasonably deserve analogous consideration. These risks becoming a gap in studies concerning the role of S100B in physiological and/or pathological conditions. In any case, with this potential problem in mind, this review will focus on the involvement of S100B in diseases of the nervous system, leaving out extra neural disorders such as obesity, diabetes, and melanoma. Indeed, the cell distribution of this protein does not offer conclusive clues to its functional role(s). In general terms, S100B, such as other calcium-binding proteins belonging to the S100 family, appears to regulate a variety of intracellular activities, interacting with different molecules located in different cell types [4]. The different functions attributed to the protein, which include the regulation of cellular calcium homeostasis and enzyme activities, interaction with the cytoskeleton, cell survival, cell differentiation, and cell proliferation, do not appear to delineate a clear, univocal intracellular role for S100B. 

In contrast, growing evidence indicates an increasingly clearer role for S100B when it is secreted—reasonably mostly by astrocytes in the nervous system—in the extracellular compartment. Extracellular S100B is regarded as interacting with target cells mainly, but not exclusively, through the muti-ligand transmembrane Receptor for Advanced Glycation End Products (RAGE), initiating intracellular signaling cascades that may result in physiological regulation at low nanomolar concentrations (“Jekyll side”) or various pathological conditions, acting as a Danger/Damage Associated Molecular Pattern (DAMP) protein at higher micromolar concentrations (“Hyde side”) [5]. RAGE is regarded as a pattern recognition receptor capable of recognizing molecules detectable in pathological conditions, such as DAMPs. The activation of the extracellular RAGE domain, in particular the V domain, which is located at the most lateral position from the plasma membrane, activates transcription factors such as NF-κB, leading to increased expression of proinflammatory cytokines [6]. Following the synthesis of the molecule, post-translational modifications induced by the interaction of extracellular S100B with its receptor are currently under active investigation to tailor an adequate modulation of these phenomena [7]. Thus, increased levels of S100B have also been shown in different biological fluids (cerebrospinal fluid, peripheral and cord blood, amniotic fluid, saliva, urine, and feces) during various pathological conditions involving the nervous system (neurodegenerative diseases such as Alzheimer’s disease (AD), Parkinson’s disease (PD), amyotrophic lateral sclerosis (ALS), multiple sclerosis (MS), traumatic and vascular acute brain injury, epilepsy, and also inflammatory bowel disease, perinatal neural disorders, glioma, and psychiatric disorders) [8,9], but also extra-neural districts (obesity and diabetes, melanoma) [3]. Although the wide spectrum of diseases in which the protein is involved reduces its specificity, levels of S100B protein in biological fluids are recognized as an important aid in monitoring the trend of the disorder, also in consequence of therapeutic approaches, and constitute a reliable, even predictive, biomarker of active distress [5]. Interestingly, in this respect, serum S100B levels have also been shown to constitute a marker of severity in COVID-19 patients [10].

In this review, while the role of S100B as a reliable biomarker is essentially generally accepted, we focus on the more recently emerging evidence individuating the S100B protein as a pathogenic factor potentially involved in processes caused by different etiologic factors and displaying different symptoms, possibly sharing aspects attributable to inflammation. This is essentially based on results obtained in experimental models of different diseases and on the significant effects obtained by modulating this protein (Figure 1). Of course, data from humans will be needed to validate S100B for clinical use. The following sections of this review will show alterations of S100B in different disorders (AD, PD, ALS, MS, traumatic and vascular acute brain injury, epilepsy, and inflammatory bowel disease), as well as indicate how alterations of the protein in experimental models of disease correlate with clinical symptoms and/or pathological parameters.

### 1.1. S100B in Alzheimer’s Disease

Astrocytes, which are known to be the main site of concentration for S100B in the central nervous system, are also known to be its main homeostatic regulator; in AD, they exert both neuroprotective and neurotoxic effects depending on the disease stage and microenvironmental factors. Essentially, based on their involvement in neuroinflammatory processes through the activation of intracellular pathways and the release of proinflammatory cytokines, they are believed to actively participate in AD pathogenic processes [11]. Astrocytic S100B has been shown to be upregulated in tissues of AD patients, and its abnormal levels, as a neurotrophic factor, have also been regarded as one possible explanation for the increased concentration of aggregates of overgrown neurites in the neuritic plaques [12,13,14,15,16]. S100B in biological fluids has been regarded as a reliable biomarker for the disease [17]. In particular, S100B levels in cerebrospinal fluid, together with other AD biomarkers such as amyloid β and phosphorylated τ, have recently been shown to have distinctive associations with higher gray matter volumes and increased glucose metabolism in key Alzheimer-related regions [18]. In fact, experiments using AD animal models indicated a significant role for the protein in AD pathogenic processes. When transgenic mice overexpressing S100B (TghuS100B mice) were crossed with the Tg2576 mouse model of AD, brain parenchymal and cerebral vascular β-amyloid (Aβ) deposits and Aβ levels were increased, accompanied by reactive astrocytosis and microgliosis and increased production of inflammatory cytokines [19]. Likewise, Aβ42 levels were significantly increased in the hippocampus and frontal cortex of transgenic mice overexpressing S100B, also exhibiting, interestingly, a sex-dependent manner [20]. On the contrary, inhibition of the protein ameliorated clinical conditions and pathological parameters in animal experimental models of the diseases. In particular, when pentamidine (PTM), an antiprotozoal drug that blocks S100B action, was administered to the mouse model of AD obtained using Aβ 1–42, reduced neuronal loss and gliosis were observed [21]. Likewise, the inhibition of astrocytic S100B synthesis by administering arundic acid (AA) reduced Aβ and amyloid plaque-associated gliosis in transgenic mice overproducing mutant amyloid precursor protein [22]. Additionally, genetic ablation of S100B (S100B knockout mice) resulted in reduced astrocytosis, microglia, dystrophic neurons, and plaques in animals generated by crossing transgenic AD model males with S100B knockout females [23]. In addition, gene polymorphisms upregulating S100B expression were interestingly shown to be associated with an increase in AD risk [24]. However, in vitro data indicating that S100B multimers act as complementary suppressors of Aβ42 oligomerization and aggregation, underpinning their potential neuroprotective role in AD, have also been reported, with the limitations of the mere in vitro approach [25]. 

### 1.2. S100B in Parkinson’s Disease

S100B has been reported to be overexpressed in the brain tissue of PD patients [26], as well as in animal PD models obtained using 1-methyl-4-phenyl-1,2,3,6-tetrahydropyridine (MPTP) [27,28]. In addition, it has been demonstrated that mice overexpressing S100B are prone to developing Parkinsonian features [29] and that the exposure of midbrain cultures to S100B has been reported to specifically alter the activity of tyrosine hydroxylase-expressing (TH^+^) dopaminergic neurons [30]. Interestingly, overnight S100B elevation has been shown to correlate with increased PD severity and sleep disruption [31], while S100B polymorphisms appear to be associated with the age of onset of PD [32]. The use of S100B as a biomarker in biological fluid for monitoring the disease has also been considered based on results obtained in PD patients [33]. 

Additionally, in the PD experimental animal model, the administration of the inhibitor of S100B activity, PTM, resulted in a significant amelioration of motor performance [34]. Likewise, the inhibition of astrocytic S100B synthesis resulting from AA administration induced the protection of dopaminergic neurons in MPTP-treated mice as an experimental PD animal model [35]. A crucial role of S100B pathogenic processes was confirmed in S100B knockout mice treated with MPTP to induce PD, where the lack of S100B expression was accompanied by amelioration of pathological parameters such as reduced loss of dopaminergic neurons, reduced microgliosis, and reduced expression of tumor necrosis factor (TNF) alpha [27]. The role of S100B in PD pathogenesis has also been recently delineated in a comprehensive review [33].

### 1.3. S100B in Amyotrophic Lateral Sclerosis

High levels of S100B protein have been described in neural tissues, especially in astrocytes, from ALS patients [36,37], although conflicting data have been reported about the reliability of S100B as a biomarker in biological fluids for this disease [38,39,40]. S100B has also been reported to be overexpressed, together with its receptor RAGE and High Mobility Group Box 1 (HMGB1) protein, another DAMP molecule binding RAGE, in the lumbar spinal cord in the SOD-G93A rodent model for ALS. Thus, a potential role for these molecules in the progression of ALS has been further proposed [41]. Interestingly, a subpopulation of “aberrant” astrocytes overexpressing S100B and its receptor RAGE has been delineated as potentially involved in neuron toxicity, at least in the SOD-G93A animal experimental model of ALS [42,43]. However, correlations between pathological parameters of ALS and S100B levels at present are not available in vivo but only in cultured astrocytic cells. The silencing of S100B in astrocytes derived from the SOD1-G93A mouse model inhibited several genes commonly overexpressed in ALS astrocytes (TNF-α, C-X-C motif chemokine, chemokine (C-C motif) ligand 6, Glial Fibrillary Acidic Protein). Consistently, in the C6 rat astrocytoma cell line, S100B was overexpressed and extracellularly released at high levels when the SOD1-G93A mutated gene, responsible for experimental ALS induction, was transfected and transiently overexpressed [43]. Taken together, these results propose a role for S100B in ALS pathogenic processes. However, this putative pathogenic role of S100B in ALS needs to be confirmed by in vivo experiments.

### 1.4. S100B in Multiple Sclerosis

Earlier data correlating S100B to MS were obtained in the late 1970s, when high levels of the protein were detected in the cerebrospinal fluid (CSF) of MS patients during the acute phase, while levels were normal in the remission phase [44]. These results were confirmed after many years, and, in addition, S100B protein was also shown to appear at high levels in the serum of MS patients at onset [45]. The occurrence of S100B protein in the biological fluids of MS patients was found to correspond to features of the nervous tissue of MS patients. In post mortem tissues from these patients, an increased expression of S100B was detected both in active demyelinating and in chronic active MS plaques [46], and, in addition, RAGE has been shown to be overexpressed in active demyelinating lesions [45]. These data proposing the involvement of the S100B protein in MS were confirmed in experimental models of the disease. Interestingly, in a rodent demyelinating model of MS (experimental autoimmune encephalomyelitis—EAE), blockade of the S100B receptor RAGE was shown to suppress demyelination [47]. Inherently, the S100B/RAGE axis has later been shown to play a crucial role in myelination processes in oligodendrocyte cultures [48]. In EAE mice, inhibition of S100B activity using PTM [49] induced amelioration of clinical scores coherently accompanied by amelioration of pathological/biomolecular parameters, thus offering in vivo evidence that S100B plays a crucial role in pathogenic processes of MS. Consistent results were obtained after inhibition of astrocytic S100B synthesis by administering AA [50]. Finally, additional results confirming that S100B inhibition in vivo, using PTM or genetic ablation, protects from EAE were later obtained [51].

### 1.5. S100B in Traumatic and Vascular Acute Neural Injury

S100B levels in biological fluids constitute a recognized clinical parameter to evaluate patients with acute brain injury [52,53,54,55,56,57,58,59,60], and a special emphasis has been recently placed on mild traumatic brain injury [61,62,63,64,65,66,67,68,69,70,71] since S100B levels have been proposed as a reliable screening tool in this pathological condition. S100B levels in human brain tissue and their direct correlation with S100B in biological fluids are difficult to evaluate. This difficulty is mainly related to the permeability of the blood-brain barrier and the presence of S100B in extra-neural districts, this latter aspect being especially relevant in multiple trauma patients [9,72,73,74]. However, reasonably high S100B levels in biological fluids are, at least in large part, due to S100B levels and its release in brain tissue. S100B serum levels have also been proposed as a biomarker to distinguish patients with primary headaches from patients with secondary headaches, depending on other brain pathological conditions [75]. It should be noted that the use of S100B levels in biological fluids in order to monitor the effects of acute brain injury, also in light of official guidelines for this purpose, constitutes a topic that has been especially addressed in recent years. In this respect, the use of S100B as a valuable biomarker has also been compared with the other frequently used astrocytic biomarker, Glial Fibrillary Acidic Protein (GFAP) [9,63,68,69]. In fact, the use of S100B as a screening tool, especially in mild traumatic brain injury, offered potential advantages over the use of GFAP; however, conclusive results have not been obtained, which is also attributed to analytical heterogeneity among laboratories, so a direct comparison across studies was unavailable. 

Additionally, in experimental rodent models of both cerebrovascular and traumatic brain injury, S100B appears to actively participate in the pathogenic processes. In both experimental animal models (middle cerebral artery occlusion (MCAO) and controlled cortical impact), high levels of S100B were detected in injured tissues [76,77,78]. Inherently, larger infarct volumes and worse neurological deficits after permanent MCAO were shown in mice overexpressing S100B (TghuS100B mice) as compared to wild-type mice [79]. Likewise, in MCAO mice, S100B treatment inhibited M2 polarization, which is regarded to be anti-inflammatory and neuroprotective, while promoting microglia M1 polarization, which is regarded to induce inflammation and neurotoxicity, with enhanced migration ability and aggravated cerebral ischemia [80]. On the contrary, the pharmacological AA-dependent inhibition of astrocytic S100B synthesis was accompanied by the prevention of brain damage and neurological deficits, as well as delayed infarct expansion, even reducing neuroinflammatory processes and motor deficits in experimental animals with intracerebral hemorrhage or subdural hematomas [81,82,83,84]. These observations confirmed a putative role played by S100B in acute brain injury pathogenic processes. Likewise, treatment with a neutralizing anti-S100B antibody in mice subjected to controlled cortical impact attenuated microglial activation, reduced lesion size, improved neuronal survival, and induced significant improvements in sensorimotor performance and memory retention as compared with control mice treated with normal IgG or vehicle [85]. In addition, the AA-induced inhibition of S100B synthesis has been shown to recover tissue damage and memory deficits produced by hypoxia-ischemia in neonatal rats [86], while in rats prone to spontaneously hypertensive stroke, AA prevented hypertension-induced stroke and inhibited the enlargement of the stroke lesion [87]. Additionally, in rats with spinal cord injury, the AA-dependent inhibition of S100B synthesis was accompanied by a reduction of secondary lesions, an improvement of motor function [88], and suppressed neuropathic pain [89]. In addition, in S100B knockout mice, where the protein was not synthesized, functional and neuropathological ameliorations were observed after traumatic brain injury [85], and similar results were observed in rats with ischemic stroke silenced for S100B using RNA interference [90].

Interestingly, it should also be noted, on the other hand, that increased dentate neurogenesis [91], as well as hippocampal synaptogenesis [92], have been reported after intraventricular administration of S100B in an experimental model of traumatic brain injury. This apparent discrepancy might be attributed to the different roles (trophic or toxic) played by the protein at different concentrations.

### 1.6. S100B in Epilepsy

Although the origin of epileptic seizures is known to be heterogeneous, many studies have been performed to detect biomarkers that can aid the diagnosis and therapy of different forms of epileptic seizures, aiming at the identification of subjects at risk of epilepsy development and monitoring the disease [93,94,95]. Among these studies, high levels of S100B protein in adults and children with different forms of epileptic seizures have been described [96,97,98,99,100,101,102,103,104,105,106,107]. In addition, it has been reported that anti-seizure therapies such as carbamazepine, oxcarbazepine, or levetiracetam were able to significantly reduce serum levels of S100B [108,109,110].

The potential involvement of S100B in the etiopathology of epilepsy has been investigated in distinct animal models. The administration to adult rats of kainic acid, which triggers seizures and neuronal loss in a manner that mirrors the neuropathology of human epilepsy, was shown to increase S100B expression in the hippocampus as a marker of activation of a definite subpopulation of astrocytes [111] and as an inflammatory cytokine [112]. Conversely, treatment with Minozac, an inhibitor of proinflammatory cytokine upregulation, was accompanied by a reduction of S100B and other indicators of glial activation, also inducing behavioral improvement [113]. Inherently, an anti-inflammatory drug such as metformin was shown to reduce S100B brain levels in a kainic-acid-induced model of epilepsy [113]. Likewise, levels of S100B secreted from hippocampal slices from a pilocarpine-induced model of epilepsy were reduced by dexamethasone treatment [114]. Similarly, rats with chronic epilepsy, when treated with the anti-epileptic agent resveratrol, had reduced S100B levels in both CSF and blood [115]. In accordance with the above-described results, also in this disorder, as in other above-mentioned diseases, the inhibitor of astrocytic S100B synthesis, AA, has been shown to produce clinical amelioration. In particular, the downregulation of neuroinflammatory parameters and astrocyte dysfunction in young rats after status epilepticus induced by Li-pilocarpine were observed [116]. In summary, studies addressing S100B as a biomarker and/or pathogenic factor of epileptic seizures, possibly as a consequence of neuroinflammatory processes proper to these disorders, have constituted an active research trend in recent years.

Intriguingly, S100B knockout mice lacking a functional S100B gene, thus not expressing the S100B protein, have been reported to be subject to earlier and more severe seizures than wild-type mice [117]. Indeed, these results do not appear to fit the bulk of the above-reported data. Peculiarities of the epileptogenic procedure used (electrical kindling of the amygdala) or unknown consequences accompanying the S100B knockout procedure might explain these apparent discrepancies.

### 1.7. S100B in Inflammatory Bowel Disease

Converging evidence indicates a possible involvement of the S100B protein in the pathogenic processes of inflammatory bowel disease (IBD), a term indicating two conditions (Crohn’s disease and ulcerative colitis) that are characterized by chronic inflammation of the gastrointestinal tract. The enteric nervous system, putatively interacting with the gut microbiota, a population of microorganisms that influences the immune system of the host during homeostasis and disease, actively participates in IBD processes [118,119,120,121,122]. S100B is present and plays a key role in enteroglial cells, which resemble astrocytes of the central nervous system, as well as in inflammatory processes [123,124]. Taken together, these data suggest the possibility that S100B may be involved in IBD pathogenic processes [125]. 

Indeed, considering S100B as a possible IBD biomarker in biological fluids, its levels were found to be lower in patients than in healthy subjects both in serum [126] and in feces [127], while the levels of S100B in biological fluids were found to be higher in all the pathological conditions investigated as compared with healthy subjects. Possible interactions with the microbiota and/or structural/functional peculiarities of the gastroenteric compartment might explain this discrepancy. 

In any case, S100B in human enteroglial cells has been shown to be overexpressed in subjects affected by IBD, to stimulate NO production, and to correlate with the gut’s inflammatory status [128]. Similarly, in the animal model of acute colitis induced by dextran sodium sulfate, high levels of S100B were described, while macroscopic and histological/biochemical assays of colonic tissues and plasma revealed a significant amelioration after treatment with PTM, the inhibitor of S100B activity [129]. PTM also prevented intestinal inflammation, oxidative stress, enteric glia activation, neuronal loss, and histological injury in intestinal mucositis induced by 5-fluorouracil, blocking an S100B-RAGE-NFkB pathway [130]. Noteworthy, a common pathway involving S100B up-regulation and Toll-like receptor (a receptor recognizing DAMPs such as S100B) stimulation has been described in the inflamed colon [131,132]. Additionally, enteric glia-derived S100B has been proposed as a putative bridge linking colonic inflammation and cancer of the colon, given its ability to interact with factors involved in both conditions (NF-κB, RAGE, and p53). Consequently, the S100B inhibitor PTM has also been proposed as a putative anti-cancer drug [133].

Finally, as above indicated, based on the consideration that a decrease in the abundance and diversity of gut microbiota-specific genera may putatively trigger IBD-initiating events [134], the possible interactions of S100B with microbiota have been recently investigated both in silico [135] and experimentally in vivo in mice [136]. Microbiota proteins putatively interacting with S100B domains were found to be reduced in IBD patients with respect to healthy subjects, also exhibiting differences in the occurrence of interacting domains. Interestingly, these in silico inferences were experimentally confirmed in mice. In fact, S100B levels were experimentally shown to correlate with microbiota biodiversity, and the correlation was significantly reduced after treatment with the S100B inhibitor PTM [136]. These data may open novel perspectives about the potential role of S100B in the gut both as a constituent of enteroglial glial cells [123], which may release it, and putatively as a constituent of food such as milk [137,138]. Thus, the protein might mediate the regulation in the intestinal microbiota, also potentially influencing IBD pathogenic processes.

### 1.8. Perspectives

Collectively, the above data delineate a scenario where the S100B protein stands at the crossroads of different pathological conditions. Its pathogenic role has been proposed in processes caused by different etiologic factors and displaying different symptoms [3,5]. However, these disorders, regardless of their origin, appear to involve processes that share aspects attributable to inflammation [139]. This view might depict S100B as an unspecific putative pathogenic factor, comprehensibly regarded with some suspicion. However, this view also presents S100B as a wide-ranging tool that may reasonably lead to unifying solutions, thus offering a promising perspective. Interestingly in this respect, the range of diseases where S100B is putatively involved in pathogenic processes, at least based on results in experimental models, is now enlarging outside the nervous system (e.g., muscular dystrophy, obesity, diabetes, ocular disorders); potentially, it is expected to parallel the cellular distribution of the protein, which is definite but significant [3,5]. Indeed, at present, the S100B protein is not the unique inflammatory molecule for which a wide therapeutic target role is putatively recognized. As examples, the prototype of DAMP proteins, HMGB1, as well as the inflammatory mediators’ resistin-like molecules, are currently regarded as interesting unifying therapeutic targets for different disorders [140,141,142]. It should also be noted in this respect that RAGE ligands, including DAMPs, have been considered potential therapeutic targets for a variety of disorders [143]. 

Thus, drugs able to efficaciously counteract S100B have been actively searched and individuated and are currently searched. Some molecules, such as AA [82], PTM [144], and TRTK12 peptide [145], have been individuated in animal models, cell cultures, or even by screening a bacteriophage random peptide display library, respectively. For PTM, an intranasal delivery system aimed at more easily reaching the nervous system protected by the blood-–brain barrier (chitosan coated niosomes) has also been proposed [133]. While the mechanism by which AA inhibits the synthesis of astrocytic S100B is still unclear, PTM and TRTK12 are known to block the interaction between S100B and the transcription factor p53. In any case, the use of AA and PTM has been widely validated in animal experimental models of different diseases, as indicated above. It is also relevant that various approaches and investigations to individuate additional, even more efficacious, molecules able to counteract S100B are currently in progress, mostly using stereological procedures [145,146,147,148]. A necessary step towards a clinical setting for S100B antagonists will be clinical trials. Clinical trials on the use of AA in neurodegenerative disorders (Alzheimer’s and Parkinson’s diseases, amyotrophic lateral sclerosis) have already been proposed and completed, although results do not appear to be available (NCT00694941, NCT00212693, NCT00403104, NCT00083421). 

In addition, owing to the recognized role of S100B as a biomarker, the clinical targeting of S100B at present may reasonably be supported by the standard hospital-based tests that measure this molecule in biological fluids. Mostly, these tests are based on the widely diffused ELISA method, and have already been tuned; numerous kits for measuring S100B in human biological fluids are commercially available. This multifaceted but unifying view of S100B as a pathogenic factor may reasonably stimulate research aimed at clarifying putatively common pathogenic processes in a variety of disorders, apparently heterogeneous for symptoms, target districts, and etiologies, but sharing the dysregulation of a multifaceted pathogenic factor as the S100B protein.

## Figures and Tables

**Figure 1 ijms-24-09605-f001:**
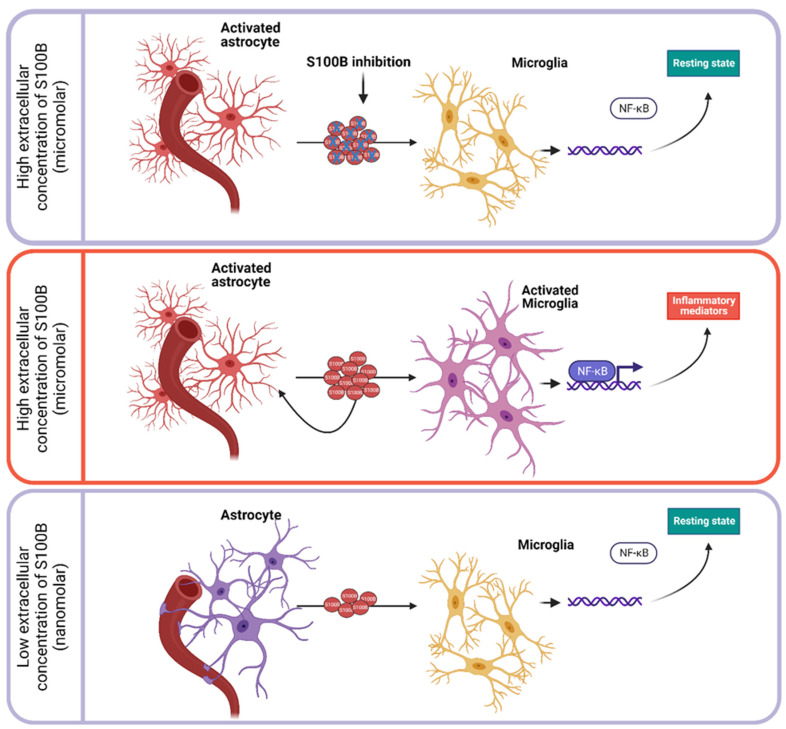
Schematic representation of effects induced by extracellular astrocytic S100B. At nanomolar concentrations, S100B does not impact the homeostasis of microglia. In contrast, when S100B reaches micromolar concentrations, microglia rapidly activate NF-κB-dependent transcription and exhibit pro-inflammatory phenotypes. Negative regulation of extracellular S100B can reprogram microglia from inflammation towards homeostasis via suppression of NF-κB signaling.

## Data Availability

Not applicable.

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
