# Peer review of "The S100B Protein: A Multifaceted Pathogenic Factor More Than a Biomarker"

_ijms, 2023, doi:10.3390/ijms24119605_

Round 1

Reviewer 1 Report

This review is in general a well-structured and written paper to summarize experimental evidence supporting biomarker values of S100B in different neurologic diseases based on its deranged levels, however, with apparent low specificity. The main drawback lies at the fact that recent similar reviews can be found in the literature, logically questioning the necessity of this one. Meaning, what are the points not covered in recent reviews but addressed here that makes this review original/needed? An example of the indicated recent similar reviews by the authors is:

Michetti F, Di Sante G, Clementi ME, Sampaolese B, Casalbore P, Volonté C, Romano Spica V, Parnigotto PP, Di Liddo R, Amadio S, Ria F. Growing role of S100B protein as a putative therapeutic target for neurological- and nonneurological-disorders. Neurosci Biobehav Rev. 2021 Aug;127:446-458. doi: 10.1016/j.neubiorev.2021.04.035. Epub 2021 May 7. PMID: 33971224.

An example by other authors: Langeh U, Singh S. Targeting S100B Protein as a Surrogate Biomarker and its Role in Various Neurological Disorders. Curr Neuropharmacol. 2021;19(2):265-277. doi: 10.2174/1570159X18666200729100427. PMID: 32727332; PMCID: PMC8033985.

By the tittle “..multifaceted therapeutic target..” readers may expect a focus on potential treatments which it is not the case. The paragraph before the last in the Introduction indicates as the focus of the review the possibility to exploit S100B as a therapeutic target, but this focus seems replaced by a more historic descriptive of data linking S100B to different neurologic diseases. Certainly some drugs are described here and there but often without specifying whether the data comes from cell culture, animal or clinical trials.

Registry numbers for four clinical trials are just listed in the section “Perspectives” with not much more detail. Perhaps a more in-depth summary of the objectives/ treatments or results (if any) together with a tabulated summary of treatment options targeting S100B for the different diseases reviewed would be more in line with what a reader would expect from this paper title. Standard S100B hospital test are also mentioned without specifying any methods or cites. Could these test serve to monitor the efficacy of the trials?

Additional improvements would be to add specific cites to the many arguments in the introduction which only provides 6 citations in 1 page and a half.

English minor: On line 68 remove double dot and the extra space after "case,". On line 70, add "or" after "diabetes". On line 329 "shearing" seems misspelled. Line 332 add "of" after "regardless"

Author Response

This review is in general a well-structured and written paper to summarize experimental evidence supporting biomarker values of S100B in different neurologic diseases based on its deranged levels, however, with apparent low specificity. The main drawback lies at the fact that recent similar reviews can be found in the literature, logically questioning the necessity of this one. Meaning, what are the points not covered in recent reviews but addressed here that makes this review original/needed? An example of the indicated recent similar reviews by the authors is:

Michetti F, Di Sante G, Clementi ME, Sampaolese B, Casalbore P, Volonté C, Romano Spica V, Parnigotto PP, Di Liddo R, Amadio S, Ria F. Growing role of S100B protein as a putative therapeutic target for neurological- and nonneurological-disorders. Neurosci Biobehav Rev. 2021 Aug;127:446-458. doi: 10.1016/j.neubiorev.2021.04.035. Epub 2021 May 7. PMID: 33971224.

An example by other authors: Langeh U, Singh S. Targeting S100B Protein as a Surrogate Biomarker and its Role in Various Neurological Disorders. Curr Neuropharmacol. 2021;19(2):265-277. doi: 10.2174/1570159X18666200729100427. PMID: 32727332; PMCID: PMC8033985.

We thank the Reviewer for this very punctual comment. The growing knowledge on S100B topic prompted us to write this manuscript including the most recent papers :36 novel papers in the period 2022/23 cited and dicusssed, of course absent in the mentioned reviews published in 2021, which also appear to be the most recent reviews addressing the topic in general terms.

   While the large majority of papers about S100B (including reviews) regard this protein as a biomarker, this review peculiarly addresses with special emphasis its possible role of pathogenic factor, which is a less trivial aspect concerning this protein.    

By the title “..multifaceted therapeutic target..” readers may expect a focus on potential treatments which is not the case. The paragraph before the last in the Introduction indicates as the focus of the review the possibility to exploit S100B as a therapeutic target, but this focus seems replaced by a more historic descriptive of data linking S100B to different neurologic diseases. Certainly some drugs are described here and there but often without specifying whether the data comes from cell culture, animal or clinical trials.

This consideration is especially welcome, and the title has been modified accordingly: “A multifaceted pathogenic factor” instead of “A multifaceted therapeutic target”, reducing the emphasis towards possible drugs affecting S100B. Also, the cited paragraph in the Introduction section has been modified accordingly.

Registry numbers for four clinical trials are just listed in the section “Perspectives” with not much more detail. Perhaps a more in-depth summary of the objectives/ treatments or results (if any) together with a tabulated summary of treatment options targeting S100B for the different diseases reviewed would be more in line with what a reader would expect from this paper title. Standard S100B hospital test are also mentioned without specifying any methods or cites. Could these test serve to monitor the efficacy of the trials?

Details of the cited clinical trials have been added in the Perspectives section. Standard S100B diagnostic tests, reasonably also used to monitor the efficacy of trials, are usually based on ELISA method, as indicated in the present version of the manuscript. At present many ELISA kits for measuring S100B are commercially available, as also indicated in the manuscript..

Additional improvements would be to add specific cites to the many arguments in the introduction which only provides 6 citations in 1 page and a half.

We have increased the number of citations in the Introduction section. However, in general terms, we chose to restrict the citations essentially to some reviews (by the way, in the present version we also cite the mentioned review by Langeh and Singh), since different achievements on this protein have been obtained by a very large number of papers published during a very large period.

Comments on the Quality of English Language

English minor: On line 68 remove double dot and the extra space after "case,". On line 70, add "or" after "diabetes". On line 329 "shearing" seems misspelled. Line 332 add "of" after "regardless"

The manuscript has been corrected accordingly.

forse è un po’ forte.

suggerisco di ammorbidire un po’ e modificare come segue:

The growing knowledge on S100B topic prompted us to write this manuscript including the most recent papers (34 novel papers in the period 2022/23 cited and discussed, of course absent in the mentioned reviews published in 2021 which also appear to be the most recent reviews addressing the topic in general terms).

Reviewer 2 Report

In the manuscript titled “The S100B Protein: A Multifaceted Therapeutic Target More Than A Biomarker”, the authors have highlighted the vital functions of the S100B protein and emerging pieces of evidence suggesting it as a potential therapeutic target.

The manuscript is well written. The review summarizes both the historical and future perspectives of S100B functions making the readers appreciate the importance of studying the protein for various pathophysiological conditions.

However, I believe a few minor issues should be addressed to improve the manuscript.

1. Although the manuscript is well organized, there are several subheadings like ‘S100B in Alzheimer’s disease’, ‘S100B in Parkinson’s disease’, and so on, which should be made into separate sections like ‘2. S100B in Alzheimer’s disease’, ‘3. S100B in Parkinson’s disease’, and so on.

Now, the entire manuscript is a part of only one section, namely, “1. Introduction”.

2. There are a few grammatical errors that need attention. For example,

-        Line 64: “….lens ,iris, ciliary…” should be “….lens, iris, ciliary…”

-        Line 71: “In general terms, S100B, like other….”

-        Line 157: “Also, in the PD…”

-        Line 260: “ethio-pathology” or etio-pathology?

-        Line 334:  “comprenhensibly” should be “comprehensively”

Please see comments above

Author Response

REVIEWER 2

In the manuscript entitled “The S100B Protein: A Multifaceted Therapeutic Target More Than A Biomarker”, the authors have highlighted the vital functions of the S100B protein and emerging pieces of evidence suggesting it as a potential therapeutic target.

The manuscript is well written. The review summarizes both the historical and future perspectives of S100B functions making the readers appreciate the importance of studying the protein for various pathophysiological conditions.

However, I believe a few minor issues should be addressed to improve the manuscript.

  1. Although the manuscript is well organized, there are several subheadings like ‘S100B in Alzheimer’s disease’, ‘S100B in Parkinson’s disease’, and so on, which should be made into separate sections like ‘2. S100B in Alzheimer’s disease’, ‘3. S100B in Parkinson’s disease’, and so on.

Now, the entire manuscript is a part of only one section, namely, “1. Introduction”.

  1. There are a few grammatical errors that need attention. For example,

-        Line 64: “….lens ,iris, ciliary…” should be “….lens, iris, ciliary…”

-        Line 71: “In general terms, S100B, like other….”

-        Line 157: “Also, in the PD…”

-        Line 260: “ethio-pathology” or etio-pathology?

-        Line 334:  “comprenhensibly” should be “comprehensively”

We want to thank the Reviewer for kind comments. The manuscript has been corrected according to the suggestions advanced

Reviewer 3 Report

The manuscript is devoted to topical issues, namely, a review of the use of the S100B protein as a biomarker for various diseases of the nervous system. The limits of applicability of diagnostics are considered in sufficient detail.

I believe that it is necessary to strengthen the main conclusion of the work in the work: “…the S100B protein can be proposed as a common therapeutic target for several different diseases…”. A biomarker and a therapeutic target are very different things. In the article, it is necessary to place emphasis on therapy in order to draw such a conclusion.

Author Response

REVIEWER 3

It is expected that authors clearly and specifically review the impact of S100B on trophic support provided by astrocytes in Alzheimer disease (AD), before endorsing the protein as therapeutic target. In current manuscript, the conclusion drawn in Fig.1 is not well support by the sited literature, hence it requires more attention.

The role of astrocytes in AD, including astrocytic S100B, has been reported with more emphasis, according to the scientific literature available. The Figure and its legend  have been been modified. In the present version the emphasis on S100B inhibitors is reduced and the legend clarifies that their action is merely based on experimental models, which support the message indicated, but not on clinical results in humans.

  1. Authors are suggested to consider reviewing the fact that there isn’t a great consistency among the findings reported from heterogenous use of research models. Therefore, it is rather advised to derive support for proposal of S100B as therapeutic target from direct clinical evidence.

The proposal of S100B as a therapeutic target has been less intensely stressed, so that it has been emendated from the title; thus, the consideration that at present the proposal is supported by experimental models, with the deriving limitations, more than from direct clinical evidence, has been stressed in the Perspectives section. As above indicated, this aspect has also been clarified in the legend of the Figure 1.

  1. In line with earlier comment, if there aren’t enough clinical evidence then it is expected to change the title of the manuscript accordingly and enforce/restrict the idea of proposing S100B as ‘therapeutic target’ rather only to discussions. Authors may consider reviewing the physiological consequences of S100B-RAGE binding in this review that are central to all.

Thank you for the useful comment. As also indicated in the response to Reviewer 1 the title has been changed avoiding the term “therapeutic target”. The S100B/RAGE interaction has been more extensively treated in the Introduction section. 

  1. Authors should consider improving the clarity of their figure.

As above indicated, the Figure and its legend  havebeen modified in order to improve their clarity

  1. Please revisit the manuscript for other format related, grammatical or typos related issues.

The manuscript has been corrected accordingly.

Comments on the Quality of English Language

A more simplified tone can be adopted, this would make the review more reader friendly and have potential to enhance the reach of the review

As suggested we improved the readability of the paper where appropriate.

Reviewer 4 Report

1.      It is expected that authors clearly and specifically review the impact of S100B on trophic support provided by astrocytes in Alzheimer disease (AD), before endorsing the protein as therapeutic target. In current manuscript, the conclusion drawn in Fig.1 is not well support by the sited literature, hence it requires more attention.

2.      Authors are suggested to consider reviewing the fact that there isn’t a great consistency among the findings reported from heterogenous use of research models. Therefore, it is rather advised to derive support for proposal of S100B as therapeutic target from direct clinical evidence.

3.      In line with earlier comment, if there aren’t enough clinical evidence then it is expected to change the title of the manuscript accordingly and enforce/restrict the idea of proposing S100B as ‘therapeutic target’ rather only to discussions. Authors may consider reviewing the physiological consequences of S100B-RAGE binding in this review that are central to all.

4.      Authors should consider improving the clarity of their figure.

5.      Please revisit the manuscript for other format related, grammatical or typos related issues.

A more simplified tone can be adopted, this would make the review more reader friendly and have potential to enhance the reach of the review.

Author Response

(The authors gave the same response as above.)

Round 2

Reviewer 1 Report

In the reply to the important question “what are the points not covered in recent reviews but addressed here that makes this review original/needed?”, the authors justify the necessity of this review to the intense research activity in the recent 1-2 years, and particularly to the original contents of 36 papers. These points of novelty/new findings should be greatly appreciated by readers but in the actual version I could not get a grasp of them.

The last paragraph of the Introduction is usually the part of the paper to highlight what are the specific points that will specifically be addressed.

When compared to the contents of the author´s recent review:

Michetti F, Di Sante G, Clementi ME, Sampaolese B, Casalbore P, Volonté C, Romano Spica V, Parnigotto PP, Di Liddo R, Amadio S, Ria F. Growing role of S100B protein as a putative therapeutic target for neurological- and nonneurological-disorders. Neurosci Biobehav Rev. 2021 Aug;127:446-458. doi: 10.1016/j.neubiorev.2021.04.035. Epub 2021 May 7. PMID: 33971224.

readers might get the impression of a shortened overview version of S100B potential role on neurologic diseases rather than having the opportunity to become updated on the new findings on S100B. When comparing the only illustration provided in this paper to the illustrations on the mentioned previous review the additions seem not clear either.

The review by Langeh and Singh which the authors claim to have added to the updated version could not be found.

Line 30, is not clear what the authors intend to say by “modulation of S100B levels..” do they refer to: altered/deranged levels?

Line 109, “shearing” should read “sharing”

Line 380 including the terms “diffused” “diffusely” should be reviewed.

The text should be reviewed to avoid repeated terms.

Spelling mistakes and word repetitions detected.

Author Response

In the reply to the important question “what are the points not covered in recent reviews but addressed here that makes this review original/needed?”, the authors justify the necessity of this review to the intense research activity in the recent 1-2 years, and particularly to the original contents of 36 papers. These points of novelty/new findings should be greatly appreciated by readers but in the actual version I could not get a grasp of them.

We thank the Reviewer for clarifying further the important question already advanced in round 1, which offer us the opportunity for clarifying further our point of view, although, of course, we absolutely respect the point of view advanced by the Reviewer.

The consideration that each review should address original points which make it needed may even be reasonable. However, this consideration might be applied to the large majority of reviews published in different journals covering different scientific topics, especially when a topic a growing, as it is the case of S100B according to PubMed records. Just as an example, High Mobility Group Box 1 protein (HMGB1), which is regarded to act as a Damage/Danger Associated Molecular Pattern (DAMP) molecule, as the S100B protein, appears in the title (and constitutes the main topic) of 44 reviews published in the period 2022/2023, often addressing the same pathological condition or addressing the topic in general terms. The opportunity for writing a review may merely spring from the consideration that a topic is in fact growing, thus deserving attention. Likewise, scientific reviews not necessarily have to stress with a special emphasis most recent papers in comparison with previous reviews, but merely cite and discuss them, when relevant, as a part of an analysis of literature, evidencing possible problems and perspectives in the field, which is the meaning of a scientific review, at least in our opinion and- we feel-also in the intent of the Journal as indicated in the “Instructions for Authors”

In any case, in the present version of the revised manuscript we especially described results obtained in most recent papers, when relevant, as suggested by the Reviewer.

The last paragraph of the Introduction is usually the part of the paper to highlight what are the specific points that will specifically be addressed.

We agree with the Reviewer with the usual meaning of the last paragraph of the Introduction section. Thus, for a better clarity, we corrected the editorial structure of the manuscript. In the present revised version, the focus of the review, which is oriented towards the role of S100B as a pathogenic factor in different disorders, is indicated in the last paragraph of the Introduction section.

When compared to the contents of the author´s recent review:

Michetti F, Di Sante G, Clementi ME, Sampaolese B, Casalbore P, Volonté C, Romano Spica V, Parnigotto PP, Di Liddo R, Amadio S, Ria F. Growing role of S100B protein as a putative therapeutic target for neurological- and nonneurological-disorders. Neurosci Biobehav Rev. 2021 Aug;127:446-458. doi: 10.1016/j.neubiorev.2021.04.035. Epub 2021 May 7. PMID: 33971224.

readers might get the impression of a shortened overview version of S100B potential role on neurologic diseases rather than having the opportunity to become updated on the new findings on S100B. When comparing the only illustration provided in this paper to the illustrations on the mentioned previous review the additions seem not clear either.

While we confirm the general considerations advanced at the point 1, which essentially answer this point, we also want to stress that the revised version of the manuscript is not oriented to consider the S100B as a therapeutic target, as in the previous review. In this respect, the title actually is “The S100B protein: a multifaceted pathogenic factor more than a biomarker” and accordingly we oriented the text. Also, the figure has been further modified in the present revised version.

The review by Langeh and Singh which the authors claim to have added to the updated version could not be found.

The review by Langeh and Singh was in fact reference n.6 in the manuscript resubmitted on May 16, and is refererence n.6 in the present revised version

Line 30, is not clear what the authors intend to say by “modulation of S100B levels..” do they refer to: altered/deranged levels?

Thank you for offering the opportunity of better clarifying what we mean. Indeed, we mean changing S100B levels: e.g., lowering using drugs, neutralizing antibodies or gene ablation, or increasing using administration of the protein or transgenic animals overexpressing S100B, according to the different procedures used by different Authors in different experimental systems. This is indicated in the last paragraphs of the Abstract section and, case by case, in the text. In the present revised version, we used another term in the Abstract section to avoid misunderstanding.

Line 109, “shearing” should read “sharing”

Thank you for showing this secretarial error. The text has been corrected accordingly

Line 380 including the terms “diffused” “diffusely” should be reviewed.

We reviewed line 380, including the terms “diffused” and “diffusely”

The text should be reviewed to avoid repeated terms.

The whole text has been reviewed in order to avoid repeated terms, and also spelling mistakes and repetitions, as requested in the following point

Comments on the Quality of English Language

Spelling mistakes and word repetitions detected.

Reviewer 3 Report

The authors have improved the article. I think that it can be accepted for publication.

Author Response

The authors have improved the article. I think that it can be accepted for publication

We thank the Reviewer for his kind comments

Reviewer 4 Report

1.      Between, line 132-133, maybe it is better to establish the consequences of increased neurotrophic activity for AD and then move on to the biomarker aspect. Need further information and rephrasing.

There are few soft errors e.g., sentence structure and grammar etc. please revisit and fix those.

Author Response

Between, line 132-133, maybe it is better to establish the consequences of increased neurotrophic activity for AD and then move on to the biomarker aspect. Need further information and rephrasing.

We want to thank the Reviewer. In the present revised version of the manuscript, we followed his suggestions, which ameliorated the clarity of the text

Round 3

Reviewer 1 Report

The modifications introduced by the authors seem satisfactory.

Author Response

We thank the Reviewer for kind comments